# An Expert System for Rotating Machine Fault Detection Using Vibration Signal Analysis

**DOI:** 10.3390/s21227587

**Published:** 2021-11-15

**Authors:** Ayaz Kafeel, Sumair Aziz, Muhammad Awais, Muhammad Attique Khan, Kamran Afaq, Sahar Ahmed Idris, Hammam Alshazly, Samih M. Mostafa

**Affiliations:** 1Eco Pack Ltd. 112, Hattar Industrial State, Haripur 7040, Pakistan; ayaz2788@gmail.com; 2Department of Electronics Engineering, University of Engineering and Technology, Taxila 47040, Pakistan; sumair.aziz@uettaxila.edu.pk; 3Department of Electrical and Computer Engineering, COMSATS University Islamabad Wah Campus, Wah 47080, Pakistan; muhammadawais@ciitwah.edu.pk; 4Department of CS, HITEC University Taxila, Taxila 47040, Pakistan; attique@ciitwah.edu.pk; 5Department of Mechanical Engineering, HITEC University Taxila, Taxila 47040, Pakistan; kamran.afaq@hitecuni.edu.pk; 6College of Industrial Engineering, King Khalid University, Abha 61421, Saudi Arabia; sahar.ahmed@live.com; 7Faculty of Computers and Information, South Valley University, Qena 83523, Egypt; Alshazly@inb.uni-luebeck.de

**Keywords:** signal analysis, empirical mode decomposition, artificial intelligence, machine faults, supervised learning, support vector machines

## Abstract

Accurate and early detection of machine faults is an important step in the preventive maintenance of industrial enterprises. It is essential to avoid unexpected downtime as well as to ensure the reliability of equipment and safety of humans. In the case of rotating machines, significant information about machine’s health and condition is present in the spectrum of its vibration signal. This work proposes a fault detection system of rotating machines using vibration signal analysis. First, a dataset of 3-dimensional vibration signals is acquired from large induction motors representing healthy and faulty states. The signal conditioning is performed using empirical mode decomposition technique. Next, multi-domain feature extraction is done to obtain various combinations of most discriminant temporal and spectral features from the denoised signals. Finally, the classification step is performed with various kernel settings of multiple classifiers including support vector machines, K-nearest neighbors, decision tree and linear discriminant analysis. The classification results demonstrate that a hybrid combination of time and spectral features, classified using support vector machines with Gaussian kernel achieves the best performance with 98.2% accuracy, 96.6% sensitivity, 100% specificity and 1.8% error rate.

## 1. Introduction

Machine faults are a major cause of unexpected downtime and production loses of industries [1]. Rotating machines constitute an integral part of most industrial equipment, especially the emerging multiport energy conversion systems, e.g., wind mills, electric vehicles and hydraulics, etc. [2]. They operate continuously under harsh conditions, becoming more vulnerable to faults. Accurate and early detection of rotary machine faults is essential to achieve system level reliability and energy efficiency. In order to achieve sustained production, most industries adopt a condition based maintenance strategy which requires continuous monitoring of machines and effective detection of faults before major breakdowns [3]. The indices of machine health are assessed by analyzing the features of various signals including voltages, currents, sound, temperature and pressure, etc. [4]. However, due to noise contribution from multiple sources, accurate machine fault detection using these signal traits is a challenging task [5].

Recently, use of vibration signals for machine fault diagnosis has got significant research interest. The underlying fact is that all electro-mechanical systems produce vibrations which characterize the dynamic behavior of machine [6]. The vibration signals acquired from a machine contain a wealth of information about its state. The vibrations of a normal/healthy machine are constant and have low-amplitude, whereas the faulty machines produce time varying vibrations. The changes in vibration spectrum can be used to identify the faulty condition of a rotatory machine. Recent advancements in artificial intelligence as well as availability of low cost vibration sensors have encouraged the researchers to investigate efficient machine fault diagnosis methods using rich vibration data.

## 2. Literature Review

The published methods of machine fault detection are mostly based on three types of machine signals, i.e., current, sound and vibration. Current from stator of the motor carries significant information about its condition. In [7], time-frequency analysis based on Hermite-S method is proposed for dynamic faults detection of brushless DC motor. A classical approach for motor’s fault detection is computing power spectral density (PSD) from the Fourier Transform (FT) of current signal. However, the FT spectrum becomes non-stationary due to variable speed of the motor. In [8], use of short time Fourier transform (STFT) is proposed to extract additional information about time based variations of frequencies in stator current signatures. In [9], the rotor fault identification of a three-phase induction motor is performed using discrete fractional Fourier transform (DFT) of stator current. DFT of time domain signal of stator current is computed at different angles to construct a characteristic matrix. Next, fractal features are extracted and extension theory is applied to identify the defect types. In [10], an approach named as complete ensemble empirical mode decomposition with adaptive noise is proposed to extract distinct intrinsic mode functions of current signal; the most discriminant among them are used to detect bearing faults. In [11], STFT is adopted to obtain spectrogram of stator current and identify BLDC motor defects. Motor current analysis is also proposed to detect mechanical and electrical faults in induction motors [12]. A deep learning approach is proposed in [13] for industrial motor fault diagnosis. At first, discrete time Hilbert transform is applied to time series signals acquired from industrial machines. Next, using the absolute value of resultant analytical signal, a textured pattern of images is constructed, which is then used to train and classify deep convolutional neural networks.

In case of fault detection methods based on acoustic signals, wavelet transform is mostly used to extract time-frequency map of signals [14]. In [15], induction motor fault detection is proposed through a combination of IMFs of sound spectrum. In [16], signal analysis in the time domain is performed using the average power of sound spectrum to detect faults in three-phase induction motors. In [17], a mobile phone-based microphone is used for the detection of rotating machine faults. The research proposed that appropriate signal processing methods could overcome the limitation of poor frequency response of mobile microphone. In [18], a combination of STFT and stacked sparse autoencoder is employed along with softmax regression to classify the faults. In another study [19], a multidimensional signal acquired from several microphones is used for the detection of gear faults.

The present research on vibration signal analysis is mainly focused on identification of most discriminant features from machine’s vibration signals and efficient classification of condition patterns. The published works in this domain can be broadly categorized as time analysis, frequency analysis and time-frequency analysis methods. Time domain analysis extracts various statistical features of vibration signal such as mean, root mean square and kurtosis, etc. In [20], cyclostationarity is used as a time domain feature to detect gear faults [20]. Although simple to compute, time domain features are noise sensitive and hence effect the reliability of results. Frequency analysis is done to identify the machine faults from the frequency domain representation of vibration signals. For this purpose various implementations of Fourier transform are used [21,22,23]. However, these frequency methods assume the input signal to be linear and stationary. Spectral kurtosis is a well known frequency domain feature, mainly used for detection of bearing faults [24]. Time-frequency analysis methods obtain the signal information in time and frequency domain simultaneously; thus extracting more powerful features for fault diagnosis. A number of time-frequency methods have been proposed which include discrete wavelet transform (DWT) [25], continuous wavelet transform (CWT) [26], wavelet packet transform and comblet transform [27], etc.

The abnormalities in vibrations of machines can occur due to several causes which can be electrical or mechanical [28]. Vibration faults due to electrical problems are generated due to flux variations around the stator or broken bar of the rotor. Mechanical causes include motor unbalance or faulty bearings, etc. [29]. In [30], the authors employed an optical method to capture the vibrations of the motor. Frequency and time domain features were extracted and fed to an artificial neural network (ANN) to detect normal and faulty states. In [31], the author utilized neural networks for the detection of online electrical faults of the motor through vibration signals. Short time Fourier transform was applied to process the vibrations and neural network was employed to detect the faults. In [32], several autoencoders are proposed to identify machine faults and results are compared with support vector machines. An embedded solution for the detection of early machine faults is proposed in [33]. Vibration signals were collected from a test rig apparatus in lab environment. Signal segmentation was performed using EMD and classification of normal state, offset pulley fault, wear fault, and cracking faults were done through k-nearest neighbors. In another work [34], vibration signals were collected from rotating machinery (motor) using AX-3DS wireless tri-axial accelerometer. Three machine states namely, normal, inner race bearing fault, outer race bearing fault were discussed in study. Signal preprocessing and segmentation was achieved using EMD, followed by feature extraction. SVM classifier was trained and testing using extracted features to distinguish different data classes.

The extracted features are used to train the classifiers to distinguish between healthy and faulty machine patterns. Some interesting works are discussed as follows. In [35], the windowed power spectral density of vibration signal is used with support vector machine (SVM) to detect the normal and faulty condition of rotatory control valve. In [36], a deep statistical feature set composed of time, frequency and time-frequency features is classified using Gaussian–Bernoulli deep Boltzmann machine. The proposed method is used to identify faults in gearbox and bearing system of rotatory machines. In [37] proposes a method for fault detection of traction motor using a filter which estimates the next healthy value from the previous values of the signal. From the difference of the original and estimated signals, various statistical features are extracted and classification is performed using artificial neural network (ANN), K-nearest neighbors (KNN), SVM and random forest. In [38], multi-kernel SVM was utilized with incremental learning to design an adaptive fault diagnosis system. In [39], fault classification of induction motor is proposed by comparing the FFT spectra of acceleration signals for healthy and broken rotor bars. Wavelet packet decomposition was applied in combination with SVM to distinguish different types of bearing faults in [40,41]. In [42], A deep learning algorithm was developed for motor fault diagnosis that also keeps in considering motor speed parameters.

This work proposes a multi-domain feature analysis approach for fault detection of induction motors. The main contributions to this work are;

First, a practical dataset of vibration signals is constructed from large industrial scale 45 KW three-phase induction motors coupled with centrifugal water pumps. To acquire signals in real time, an industry standard sensor, i.e., Beanscape tri-axis acceleromter is used.The proposed method employed a data-driven approach in signal preprocessing step using Empirical mode decomposition technique.While most of the published works are based on using only a single class of features, i.e., time, frequency or time-frequency features constructed from STFT, this work proposes to use multi-class feature vectors consisting of several combinations of time domain and frequency domain features. A detailed analysis is done to study the discriminative properties of a large pool of such combinations. The most promising feature combinations resulting in high classification performance are then presented.Classification is performed using various classifiers with multiple kernel settings.

The rest of the article is organized as follows. Section 3 presents the pipeline of proposed fault detection approach, discussing all the main computational steps. Section 4 presents comprehensive performance analysis of the proposed approach. Finally, Section 5 concludes this article by giving insights into future contributions.

## 3. Materials and Methods

Figure 1 demonstrates the pipeline of proposed machine fault detection approach. The vibration signal is acquired from the machine under test using an acceleration sensor. The noise corrupted signal is preprocessed using EMD technique. Afterwards, multi domain features are extracted and fused together to obtain different combinations having high discriminating capabilities. Finally, classification is done using a range of classifiers with different settings. These steps are discussed as follows.

### 3.1. Data Acquisition

The first and foremost step of any machine learning task is dataset collection. For this work, a self-collected dataset is used consisting of physically acquired vibration signals from induction motors of large industrial enterprise. For this study, 45 KW three-phase induction motors coupled with centrifugal water pumps were used. Signal acquisition is performed using Beanscape tri-axis wireless accelerometer, a widely used vibration sensor for a variety of industrial applications. The accelerometer is mounted at various positions on faulty and normally running motors as shown in Figure 2. The accelerometer captures individual signals for vibrations along X,Y and Z axis, which are then combined into a single time domain composite signal. For supervised binary classification, the data is labelled as Normal and Faulty motors; the latter having bearing and alignment faults. The sampling frequency of the sensor was set to 1000 Hz as provided by sensor specifications. Table 1 provides a summary of acquired dataset. A total of 103 signals are collected from normal motors, whereas 117 signals are collected from faulty motors. The duration of each signal is 5 s.

### 3.2. Preprocessing

The accelerometer provides three independent channels of time domain vibration signals corresponding to *x*, *y* and *z* axis. These channels are combined to form a single time domain signal using the equation,
(1)$$S\left(t\right)=\sqrt{x{\left(t\right)}^{2}+y{\left(t\right)}^{2}+z{\left(t\right)}^{2}}$$

The signal S(t) is then normalized by dividing each sample by the maximum amplitude value. Figure 3 shows the raw signals acquired from normal and faulty motors. It can be observed that faulty motors exhibit amplitude spikes in their vibrations which shows the presence of high frequency components in the signal spectrum.

#### Empirical Mode Decomposition

The fault indicator proposed in this work is based on the observation that in the presence of faults, the spectral components of vibration signal increased as compared to healthy spectrum. Moreover, the acquired signal is also corrupted by noise and redundant information. Therefore, in the next step, the normalized signal is subjected empirical mode decomposition, an iterative technique which decomposes a raw signal into its time domain sub-components called Intrinsic Mode Functions (IMFs) [43,44,45]. The output of EMD remains same in the spectrum. An IMF characterizes the oscillatory mode implanted in the signal and it must satisfy following two properties.

The number of maxima and minima must differ atmost by 1The mean of IMF is zero

IMF decomposition of a signal is obtained by a process known as “sifting” which is performed with the following steps.

Identify all local minima and maxima of the input signal x(t)Create the upper and lower envelope of all local minima and maxima by using cubic-spline methodDesignate the mean of upper and lower envelopes as $m_{1}$Calculate $h_{1}=x\left(t\right)-m_{1}$ as the first componentIf $h_{1}$ is an IMF, take it as first IMF of x(t). Else, take $h_{1}$ is a proto-IMF and name it as $h_{11}$. Take $h_{11}$ as the original signal and repeat steps 1–4 until $h_{1k}$ is an IMF, and designate it as $c_{1}=h_{1k}$, where *k* indicates the number of iterations to produce an IMF.Obtain residum $r_{1}=x\left(t\right)-c_{1}$Treat $r_{1}$ as the original signal and apply steps 1–6 to obtain other IMFs $c_{2},c_{3},\cdots ,c_{n}$ as follows:
$$\begin{array}{ccc} r_{1}-c_{1} & = & r_{2}\\ \vdots \\ r_{n-1}-c_{n} & = & r_{n} \end{array}$$The decomposition process can be stopped when $r_{n}$ becomes a monotonic function. However, only few IMFs have physical meaning for most practical purposes. At the end of EMD, it gives a signal of the form
(2)$$x\left(t\right)=\sum_{i=1}^{n}c_{i}\left(t\right)+r_{n},$$
where x(t) is decomposed into *n* IMFs and a residue $r_{n}$. Figure 4a,b illustrate few IMFs extracted from normal and faulty signals of machines. High frequency components can be observed in lower IMFs. In this work, total 10 IMFs are extracted. It was conceived experimentally that first IMF (IMF1) contained noisy elements and redundant components. Therefore, this component was rejected. The remaining nine IMFs and residual signal were added together to construct the preprocessed signal.

### 3.3. Feature Extraction

Feature extraction is a crucial step in machine learning and pattern recognition frameworks. One type of feature is never adequate to extract all hidden characteristic information from the signals of different classes. In this work, different combinations of time, and frequency domain features are systematically tested to find out the best performing combination with the highest classification accuracy and lowest feature dimensions.

#### 3.3.1. Temporal Features

Temporal features define various statistical descriptors extracted from the time domain representation of vibration signal [46]. This work uses a number of classical time domain statistical features such as mean, standard deviation, root mean square (RMS) and signal energy to obtain differences between one vibration signal and another. In addition, due to non stationary nature of vibration signal of faulty machines, more advanced statistical features such as skewness and kurtosis are also investigated in this work. These features computed using the probability density function (PDF) of the time domain signal. Since the PDF of vibration signal changes with change in condition of machine bearing, thus skewness and kurtosis also change. The kurtosis quantifies the peak value of the PDF of the signal, whereas the skewness quantifies the asymmetry behavior of PDF. Studies show that kurtosis for a normal machine machine is approximately three and its skewness value is approximately zero [47]. In case of PDF changes due to a machine fault, the kurtosis value increases and skewness value becomes positive or negative.

#### 3.3.2. Spectral Features

Spectral features are extracted from the frequency domain representation of a signal. In order to convert the time domain signal in to frequency domain, FFT is a common method of analysis which obtains the dominant frequency of the repetitive impulse period of certain machine faults. Several classical spectral features are used in this work and obtained from amplitude spectrum of vibration signal. These features include mean frequency, median frequency and standard deviation. In addition, various advanced spectral features have also been investigated in this work. These include spectral kurtosis, spectral centroid, spectral flux, spectral roll-off, spectral flatness, spectral crest, spectral decrease, spectral slope and spectral spread. Table 2 lists all features used in this work along with their acronyms.

#### 3.3.3. Hybrid Features

In this work, the hybrid features are defined as the feature vectors consisting of various combination of temporal and spectral features.

### 3.4. Classification

In the final step, the extracted time and spectral features are fused together in different combinations and applied to a set of classifiers. In this study, a 10-folds cross-validation scheme is adopted to train/test the classification models. In this scheme, the dataset is divided into ten equal folds. In each iteration, one fold is employed for testing, and the remaining nine folds are used to train the model. This procedure is repeated 10 times, and the final performance is computed by taking an average of all runs. As shown in Table 1, the data comprises of 103 normal, 117 faulty and a total of 220 signal observations. Each observation is composed of 5000 signal samples. In each iteration of 10-fold cross-validation, 22 observations (10 Normal and 12 Faulty signals) were used for testing, and the remaining 198 (93 Normal and 105 Faulty) were employed for training the classification model. This process is iterated 10 times, and the model is tested and trained on all observations. This scheme is more preferable for a dataset of comparatively small size.

The classification is performed using different kernel settings of SVM, linear discriminant analysis (LDA), decision tree (DT) and k-nearest neighbours (KNN) classifiers.

## 4. Performance Analysis

### 4.1. Feature Analysis

In order to find the best describing feature set combination with lowest dimensions, we performed experimentation with two base line classifiers, i.e., support vector machine with Quadratic kernel (SVM-Q), and k-nearest neighbors with weighted kernel (KNN-W). The performance of these classifiers on different feature sets is demonstrated in Table 3. The feature sets F1-F3 are composed of single domain features, i.e., time or frequency, whereas F4-F7 are hybrid feature sets consisting of a combination of time as well as frequency features. The SVM-Q achieves an accuracy of 90.5%, 94.1%, and 94.1% for time, frequency and spectral features respectively. Among the various feature sets that we tried, the set F4 achieves the best classification performance using SVM-Q classifier, achieving an accuracy of 98.2%. This best combination has a feature size of 13. A full combination of all features, i.e., F7 has 96.5% accuracy however, in this case the feature dimensions becomes considerably large, i.e., 21 features.

In pattern recognition problems, a relationship between different feature classes can be efficiently illustrated using a scatter plot. Best features are identified as those having maximum inter-class difference, i.e., the means of both classes lie at maximum distance from each other in scatter plot, whereas intra-class difference is minimum for the same feature. As an example, Figure 5a shows the predictions of SVM-Q classifier on joint feature vector of temporal mean and standard deviation. Similarly, Figure 5b shows the SVM-Q predictions on frequency standard deviation and skewness features. In this way, classifier predictions were visually analyzed to obtain promising feature vector combinations for classification.

### 4.2. Classification Performance

In the next step, best feature set of Table 3, i.e., F4 comprising of time and frequency features is selected for classification using a range of classifiers namely, Linear Discriminant Analysis (LDA), Decision Tree (DT), KNN with K=10 (KNN-M), KNN with K=1 (KNN-F), KNN with the weighted kernel (KNN-W), SVM with the quadratic kernel (SVM-Q), SVM with linear kernel (SVM-L), SVM with cubic kernel (SVM-C), and SVM with Gaussian Kernel (SVM-G). Table 4 shows the performance of these classifiers in terms of standard performance metrics of specificity, sensitivity, accuracy and error. SVM-Q achieves best classification performance with 98.2% accuracy, 96.6% sensitivity, 100.0% specificity and 1.8% error rate. These results are also graphically illustrated in Figure 6. Most of the classifiers used in this work achieve an accuracy above 90% which shows the validity of proposed approach.

Figure 7 shows the confusion matrix of classification performance with SVM-G classifier. Out of 117 faulty vibration signals, only 4 are misclassified as Normal and the remaining 113 signal are correctly predicted. Moreover, all 103 signals acquired from normal/healthy motors are correctly predicted as normal by the classifier.

Figure 8 illustrates the confusion matrix information in term of sensitivity and specificity. Faulty class achieves 96.6% sensitivity, whereas the normal class achieves 100% specificity.

## 5. Conclusions and Future Work

Early and accurate machine fault detection plays an important role to ensure the productivity and economic stability of industrial enterprises. In most of emerging multiport energy conversion systems, critical functions are performed by rotating machines especially motors. The vibration signals contain a wealth of information about the health and state of the machine. However, due to time varying nature of vibration signals, using them for accurate detection of machine faults is not a trivial tasks. This calls for extraction of powerful features from vibration data and selection of appropriate classification methods. In this work, an approach is proposed for fault identification of large industrial motors. This work has three main contributions.

Since, the reliability of any system based on machine learning depends upon effectiveness of collected dataset. In this work, the vibration signal dataset is constructed from real-time, practical industrial setup rather than using data collected from laboratory environment.In order to remove the noise contribution from practical sources, an efficient signal conditioning approach is proposed based on Empirical mode decomposition.While most of the published works in this domain are concentrated on using a single class of features of fault detection, this work is based on an approach based on hybrid features. we systematically analyzed the performance of different combinations of time and frequency domain features using a range of classifiers with multiple settings.

The proposed approach can be applied to any industrial setup for real time detection of machine faults using vibration analysis. In future work, we aim to extend this work to perform multi-class identification of individual faults of machines. Moreover, increasing the number of data samples and feature reduction methods can also improve the performance with reasonable complexity.

## Figures and Tables

**Figure 1 sensors-21-07587-f001:**
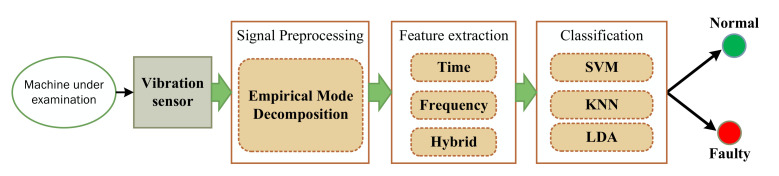
Pipeline of proposed machine fault detection approach.

**Figure 2 sensors-21-07587-f002:**
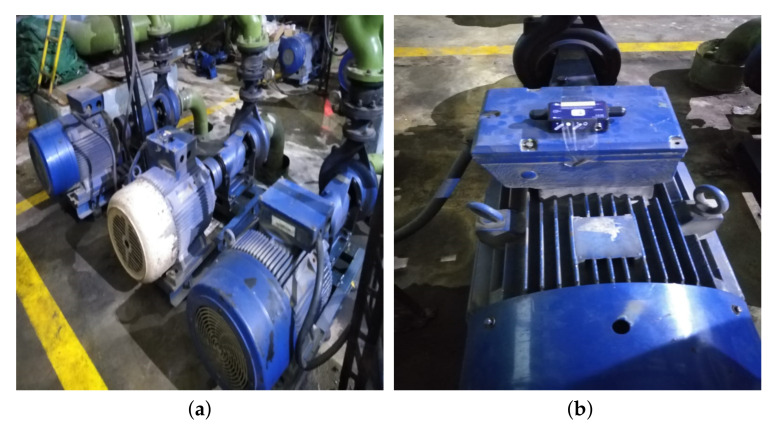
Data Acquisition Setup. (**a**) Motor Assembly. (**b**) Accelerometer Placement.

**Figure 3 sensors-21-07587-f003:**
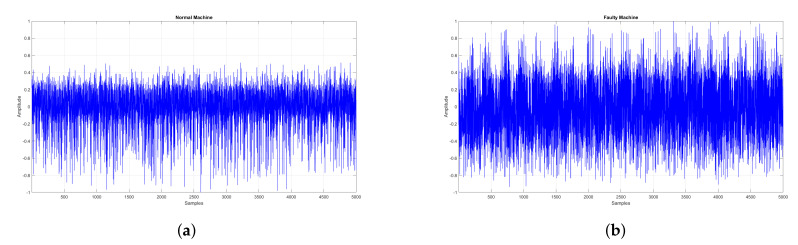
Time Domain Representation of Vibration signal for Normal and Faulty Motor. (**a**) Normal Motor. (**b**) Faulty Motor.

**Figure 4 sensors-21-07587-f004:**
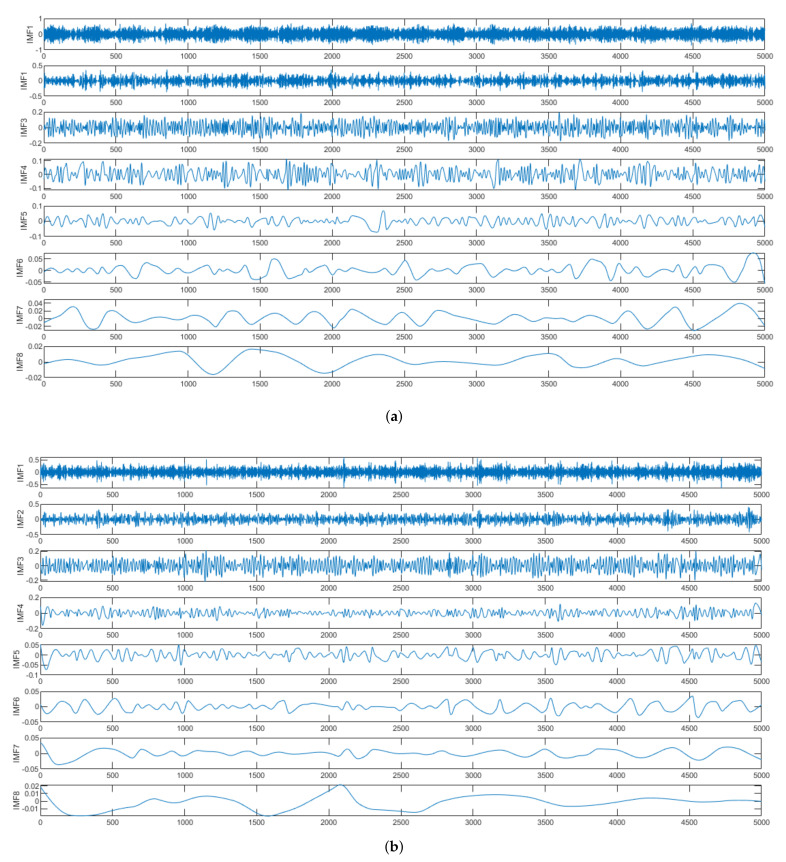
IMFs of vibration signals of Normal and Faulty machines. (**a**) Normal Motor. (**b**) Faulty Motor.

**Figure 5 sensors-21-07587-f005:**
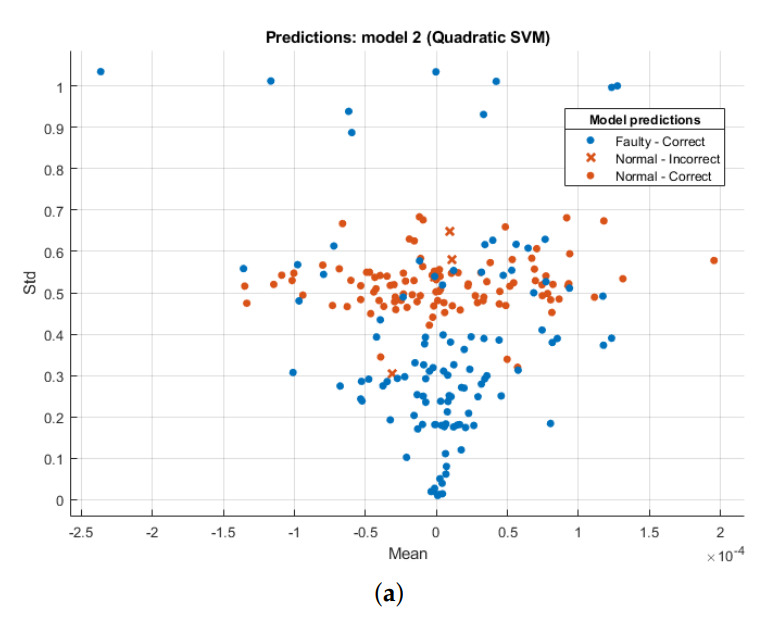
Model Predictions of SVM-Q Classifiers for various combinations of temporal and frequency domain features. (**a**) Temporal Mean and Standard Deviation. (**b**) Frequency standard deviation and skewness.

**Figure 6 sensors-21-07587-f006:**
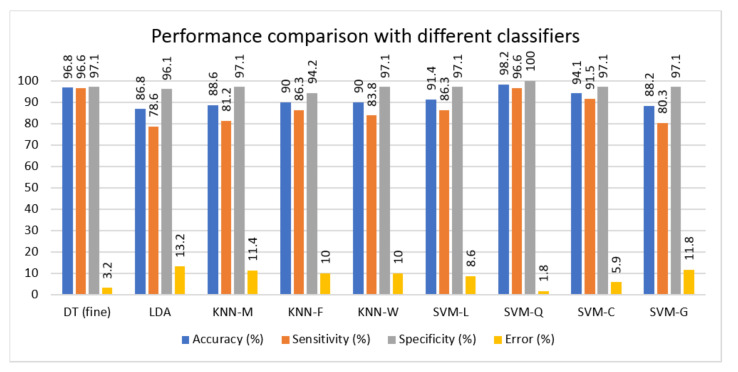
Performance comparison of different classifiers in terms of accuracy, sensitivity, specificity, and error.

**Figure 7 sensors-21-07587-f007:**
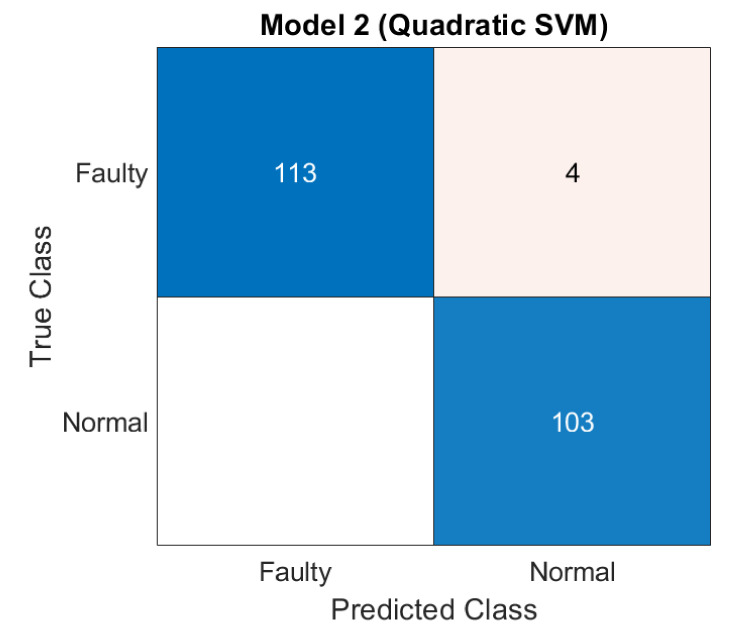
Confusion Matrix of classification using SVM-Q.

**Figure 8 sensors-21-07587-f008:**
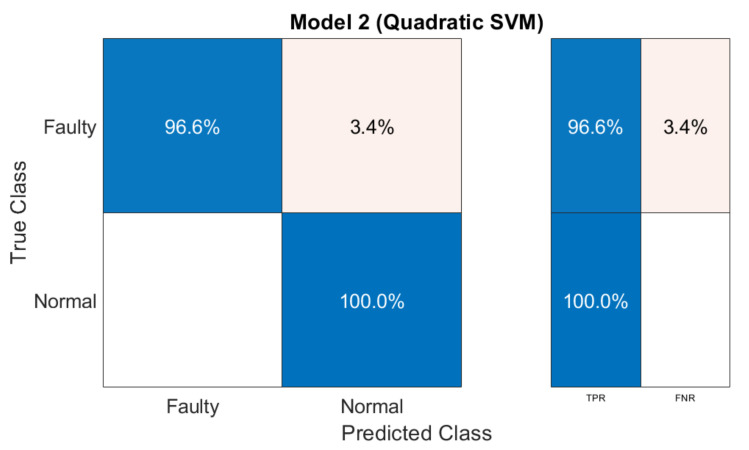
Confusion Matrix in terms of sensitivity and specificity.

**Table 1 sensors-21-07587-t001:** Details of the acquired dataset.

Data Class Name	Signals	Duration	Signal Size
Normal	103	8.5 min	5 s
Faulty	117	9.75 min	5 s
Total	220	18.25 min	-

**Table 2 sensors-21-07587-t002:** List of Features investigated for classification.

Time Domain		Frequency Domain	
**Feature**	**Acronym**	**Feature**	**Acronym**
Mean	M	Mean Frequency	FM
Standard Deviation	SD	Frequency Standard Deviation	FSD
Skewness	SK	Skewness of Frequency	FSK
Kurtosis	KR	Kurtosis of Frequency	FKR
Peak to Peak	PP	Band Power	BPWR
Root Mean Square	RMS	Median Frequency	FMED
Energy	E	Spectral Centroid	SC
		Spectral Flux	SF
		Spectral Roll Off	SRO
		Spectral Flatness	SFL
		Spectral Crest	SCR
		Spectral Decrease	SDEC
		Spectral Slope	SSL
		Spectral Spread	SS

**Table 3 sensors-21-07587-t003:** Performance evaluation of different feature sets combination with several classifiers.

Feature Set	Size	Feature Class	Features in the Set	Accuracy (%)	
				SVM-Q	KNN = W
F1	7	Temporal	M,SD,SK,KR,PP,RMS,E	90.5	90.0
F2	6	Frequency	FM,FSD,FSK,FKR,BPWR,FMED	94.1	93.6
F3	8	Frequency	SC,SF,SRO,SFL,SCR,SDEC,SSL,SS	94.1	92.3
**F4**	**13**	**Hybrid**	**M,SD,SK,KR,PP,RMS,E** **FM,FSD,FSK,FKR,BPWR,FMED**	**98.2**	**90.0**
F5	14	Hybrid	FM, FSD, FSK, FKR,BPWR,FMED, SC,SF,RO,SFL,SCR,SDEC,SL,SS	95	91.4
F6	15	Hybrid	M, SD, SKW,KR,PP,RMS,E, SC,SF,SRO,SFL,SCR,SDEC,SSL,SS	91.4	91.4
F7	21	Hybrid	M,SD,SK,KR,PP,RMS,E FM,FSD,FSK,FKR,BPWR,FMED SC,SF,SRO,SFL,SCR,SDEC,SSL,SS	96.5	90.5

**Table 4 sensors-21-07587-t004:** Performance of feature set **F4** with a range of classification methods.

Classifier	Accuracy (%)	Sensitivity (%)	Specificity (%)	Error (%)
DT (fine)	96.8	96.6	97.1	3.2
LDA	86.8	78.6	96.1	13.2
KNN-M	88.6	81.2	97.1	11.4
KNN-F	90	86.3	94.2	10
KNN-W	90	83.8	97.1	10
SVM-L	91.4	86.3	97.1	8.6
**SVM-Q**	**98.2**	**96.6**	**100**	1.8
SVM-C	94.1	91.5	97.1	5.9
SVM-G	88.2	80.3	97.1	11.8

## Data Availability

The dataset and code of this work can be provided on request.

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
