# Peer review of "An Expert System for Rotating Machine Fault Detection Using Vibration Signal Analysis"

_sensors, 2021, doi:10.3390/s21227587_

Round 1
Reviewer 1 Report
The authors designed an expert system for machine fault diagnosis based on EMD and machine learning. The novelty of the manuscript is not very high. Suggestions and comments are as follow:
1) Is the EMD used to denoise the raw signal by deleting IMF1? Please give a clear illustration which IMFs are used after EMD.
2) The illustration of Figure 8 should after the illustration of Figure 7.
3) It would be better to combine section 1 and section 2.
4) Equation 1 only can obtain the amplitude of the combined signal. How to obtain the phase of the combined signal?
5) Please show the entire signal in Figure 3 a. It looks like the amplitude of the normal signal is obviously higher than the fault signal in Figure 3. It is abnormal from my experience.
6) To improve the novelty of the manuscript, I suggest the authors do more work about the combination of the features to test which temporal features and frequency features are the most important.
7) In section 2, the authors know that spectral kurtosis are widely used for bearing fault diagnosis but why the SK is not used as a feature?
8) Whether is it possible to classify the fault type between bearing fault and misalignment?
Author Response
File attached

Reviewer 2 Report
The structure of the paper is fair; however, I have found errors and imprecise statements. I don't know whether it comes from poor English technical language knowledge or misunderstanding some terms and topics connected to (digital) signal processing (DSP). I have pointed out some examples:
I strongly insist on not using the often overused term "intelligent" (e.g. "intelligent fault detection system," "intelligent filter"), especially in scientific papers. It's applicable rather only in advertisements.
It is unknown what the Authors mean, stating (line 70-71) "... preprocessing of data is performed using Hilbert transform, followed by conversion of preprocessed signals to texture images" as well as in (lines 73-74) "... wavelet transform is mostly used to extract signal energy features" (what features has the energy?). It's imprecise.
The are no multichannel signals (line 83), "multidimensional signal" would sound better.
FFT (Fast Fourier transform ) is just an algorithm of calculating Discrete Fourier Transform, not a different method (lines 93-94).
"FFT spectrums" - should be "FFT spectra"
The statement "Due to nonlinear nature of vibration signals..." is wrong. It depends on the researcher whether a phenomenon can be treated as linear or nonlinear. It's just a question of choosing the appropriate model.
All the variables in Eq. 1 are time-dependent. Initially, the signal is denoted as s (should be s(t)), next x(t). Youshould correct it.
The "frequency features" and "spectral features" belongs to the same class (Figure 1). It means the same. Signals can be analyzed in the time domain, frequency domain, or in combined space – the time-frequency plane (Short-time Fourier transform, wavelet transform). All in all, the data flow presented in Fig. 1 is wrong and, consequently, Table 2.
What decided of setting the sampling frequency (fs) to 1000Hz and the length of signals to 5s (you've investigated a series of 5000 samples)? Have you any knowledge of what would justify setting the fs to 1kHz?
For the sake of clarity, the investigated features should be defined or mentioned.
Nothing is known about the acquired data set. How it was splitted into the training and the testing sets. Have you any data used for validation (acquired from the different machines (both good and faulty)?
What decided of selecting the elements of feature vector (section 3.3)?
It was not proved that the approach presented in the paper could be applied in real-time because there is no information about hardware and software implementation.
Author Response
Response Sheet added

Reviewer 3 Report
This paper proposes an expert system for fault detection of rotating machines based on the analysis of vibrations.
1.- Abstract: “This work proposes an intelligent.” Intelligent and expert are different words with different meanings. Please clarify.
2.- The methods applied in this paper are well described in the technical bibliography, even in the area analyzed in this paper, so I cannot see any new contribution and the authors do not report the possible contributions.
Author Response
Response sheet attached

Reviewer 4 Report
Reviewed article concerns an expert system for rotating machine fault detection using vibration signal analysis and is write in accordance with generally accepted standards of the scientific works. After careful reading of the submitted text there are some remarks that should be taken into consideration by the Authors to improve reviewed text.
- I suggest providing more precise information about used experimental and measurement positions.
- Presented study widely covers defined scientific problem and with experimental investigations provides proper background for given conclusions, however deeper scientific consideration of obtained results referred to the basic phenomena in analyzed processes should be given.
- I suggest also to give wider description of potential use of presented findings in scientific research as well as in industrial practice.
- The strengths and limitations of the obtained results and applied methods should be clearly described.
- I suggest providing the main conclusions as numbered sentences and refer to specific values (results of analysis) as well as basic phenomena that cause described results.
- The conclusions should highlight the novelty and contribution to the state of the knowledge in given area.
Author Response
Response sheet attached

Round 2
Reviewer 1 Report
The manuscript has been revised according my suggestions. It can be pubished at present form.
Author Response
Response sheet attached

Reviewer 2 Report
1. The authors have made the necessary corrections. However, not all are clear. For example, you wrote (line 130) "Due to non-stationary nature of vibration signals ... ". It's hard to point a "nature" of the signal for it is just an abstract term. It depends on the researcher which model describing a phenomenon is chosen.
3. What are "large industrial scale motors" (line 135) and "standard vibration sensor" (l. 136)? . It does explain anything. Just provide the information about the type/model of the motor and vibration sensor. Actually, it was mentioned in the text (Section 3.1).
Author Response
Response sheet attached

Reviewer 3 Report
The authors have replied all my concerns. There is a typo in Table 4
Author Response
Response sheet attached
